# Assessment of global health risk of antibiotic resistance genes

Zhenyan Zhang[1,6], Qi Zhang[1,6], Tingzhang Wang[2,6], Nuohan Xu[1], Tao Lu ⓘ [1], Wenjie Hong[2], Josep Penuelas ⓘ [3,4], Michael Gillings ⓘ [5], Meixia Wang[2], Wenwen Gao[2] & Haifeng Qian ⓘ [1✉]

Antibiotic resistance genes (ARGs) have accelerated microbial threats to human health in the last decade. Many genes can confer resistance, but evaluating the relative health risks of ARGs is complex. Factors such as the abundance, propensity for lateral transmission and ability of ARGs to be expressed in pathogens are all important. Here, an analysis at the metagenomic level from various habitats (6 types of habitats, 4572 samples) detects 2561 ARGs that collectively conferred resistance to 24 classes of antibiotics. We quantitatively evaluate the health risk to humans, defined as the risk that ARGs will confound the clinical treatment for pathogens, of these 2561 ARGs by integrating human accessibility, mobility, pathogenicity and clinical availability. Our results demonstrate that 23.78% of the ARGs pose a health risk, especially those which confer multidrug resistance. We also calculate the antibiotic resistance risks of all samples in four main habitats, and with machine learning, successfully map the antibiotic resistance threats in global marine habitats with over 75% accuracy. Our novel method for quantitatively surveilling the health risk of ARGs will help to manage one of the most important threats to human and animal health.

---

[1] College of Environment, Zhejiang University of Technology, 310032 Hangzhou, P. R. China. [2] Key Laboratory of Microbial Technology and Bioinformatics of Zhejiang Province, Hangzhou 310012, P. R. China. [3] CSIC, Global Ecology Unit CREAF-CSIC-UAB, Bellaterra, 08193 Barcelona, Catalonia, Spain. [4] CREAF, Campus Universitat Autònoma de Barcelona, Cerdanyola del Vallès, 08193 Barcelona, Catalonia, Spain. [5] ARC Centre of Excellence in Synthetic Biology, Faculty of Science and Engineering, Macquarie University, Sydney, NSW 2109, Australia. [6] These authors contributed equally: Zhenyan Zhang, Qi Zhang, Tingzhang Wang. ✉email: hfqian@zjut.edu.cn

Antibiotic resistance is an increasing global threat to human health and to the clinical treatment of disease[1]. Antibiotic resistance genes (ARGs) have been detected in the last decade in all environments, including natural[2–4], engineered[5–8], and clinical[9,10] habitats. Anthropogenic activities, including the clinical use of antibiotics, are widely regarded as the main drivers of the dissemination of ARGs[6,11,12].

ARGs, however, do not arise from current human activities. They existed prior to the antibiotic era, having been detected in permafrost[13] and human paleofeces[14]. Anthropogenic activities instead drive the selection of genes from environmental and cellular sources, and these can be subsequently co-opted to confer antibiotic resistance. These genes originally had a range of environmental functions, such as liberating phosphorus from phosphate[15], or encoding efflux pumps. Diverse genes can potentially confer resistance to antibiotics, so we need to determine whether genes bioinformatically identified as ARGs pose a health risk to human health, and to do so from multiple perspectives[16–18].

The first consideration is the potential for the transmission of ARGs from the environment to bacteria in humans[16], defined here as "human accessibility". ARGs can transfer from the environment to humans, via their bacterial hosts, and then have a negative impact on human health[6,19]. Mass sampling and the development of metagenome-assembled genomes (MAGs) have allowed us to understand the distribution of ARGs and their hosts in specific habitats[4–7,9]. Danko et al. (2021) assembled the first atlas of urban metagenomics using 4728 metagenomic samples from mass-transit systems in 60 cities[7]. Understanding the distribution and dissemination of ARGs in global habitats from the perspective of global health is important, especially from environmental compartments to humans[20]. The second consideration is the potential for the transmission of ARGs from environmental bacteria to pathogens[16–18], here defined as "mobility" and "human pathogenicity" of the ARGs. From this perspective, only ARGs in pathogenic hosts that can infect humans pose an elevated risk to human health[16]. ARGs frequently move by horizontal gene transfer (HGT) from nonpathogens to pathogens, and this activity has been a major driver in the evolution of pathogens resistant to antibiotics[12,21,22]. The third perspective is the current clinical use of antibiotics[17], here defined as the "clinical availability" of the ARGs. The use of antibiotics has increased globally in recent years[23], and some new antibiotics have been developed for clinical use[24,25]. In contrast, some antibiotics are now rarely used. Considering clinical relevance in evaluating the risk of ARGs is thus necessary.

In this work, based on the three perspectives outlined above, we use 4572 metagenomic datasets to reveal the distribution and dissemination of 2561 ARGs as well as their hosts in global habitats. In the next step, we carry out a framework to quantitatively evaluate the health risk of each ARG and sample by considering the four indicators (human accessibility, mobility, human pathogenicity, and clinical availability). There are 23.78% of the ARGs pose a health risk, especially those which confer multidrug resistance. Finally, we predict and map the antibiotic resistance threats in marine habitats using machine learning.

## Results

**Global patterns of ARG distribution**. We used a set of 4572 metagenomic samples to illustrate the global patterns of ARG distribution (Supplementary Data 1). These samples were collected from six types of habitats: air, aquatic, terrestrial, engineered, humans and other hosts (Fig. 1a and Supplementary Data 1). From these samples, we identified a total of 2561 ARGs that conferred resistance to 24 drug classes of antibiotics based on

the Comprehensive Antibiotic Research Database (CARD). Of these, 2401 were genes conferring resistance to only one drug class, and 160 conferred resistances to multiple drug classes (Supplementary Data 2). Twenty-five ARGs were found in more than 75% samples, however, the frequency of most ARGs (2313/2561) were <10% (Supplementary Fig. 1a). On the other hand, nearly half of 2561 ARGs were commonly shared by diverse habitats (Supplementary Fig. 1b), especially genes conferring resistance to widely used antibiotics[26] like aminoglycosides, tetracyclines, and beta-lactams (Supplementary Fig. 1c). These results implied that anthropogenic activities, like the use of antibiotics, were critical for dissemination of ARGs globally.

We examined the abundance and composition of ARGs in diverse sub-habitats at a global scale. The human-associated habitats, including the digestive system and skin, had the highest abundances of ARGs (Fig. 1b). Built environments, mainly including urban subways, also had considerable abundances of ARGs, confirming these as hotspots of ARGs[7]. Genes conferring resistance to tetracyclines and aminoglycosides, two widely used antibiotics in the clinic[26], dominated in digestive system and skin, respectively, while ARGs conferring multidrug resistance had a high proportion in built environments (Fig. 1c). Although geographic factors like latitude were reported to influence the abundance of the ARGs[4], confirmed in this study (Supplementary Fig. 2), anthropogenic activities are also critical for dissemination of ARGs. To further determine the impacts of anthropogenic activities on the dissemination and abundance of ARGs, we calculated population densities at each sample site.

**Effect of anthropogenic activities on the dissemination of ARGs**. Sampling sites were clustered into two groups according to their general population density, one with high-intensity activity (>58 people/km², the global average population density[27]), and the other with low population density. Regions of high-intensity activity had significantly higher total abundances of ARGs and genes conferring resistance to specific classes ($p < 0.001$, two-tailed Welch's $t$-test; Fig. 1d and Supplementary Fig 3). A total of 671 ARGs were specifically detected in environments with high-intensity activity (Fig. 1e). For ARGs shared between high- and low-intensity environments, 715 were significantly more abundant in high-intensity environments ($p < 0.05$, two-tailed Welch's $t$-test; Fig. 1f) and most of these conferred beta-lactam and multidrug resistance (Supplementary Fig. 4).

Thirty-four ARGs were specific to low-intensity environments, mainly annotated as beta-lactam resistance genes (adjust $p > 0.05$, two-tailed Welch's $t$-test; Fig. 1e and Supplementary Fig. 4). The distributions of 1102 ARGs were not significantly influenced by anthropogenic activities (adjust $p > 0.05$, two-tailed Welch's $t$-test; Supplementary Data 3). These genes are probably not resistance genes that will affect human health and likely perform different functions for their bacterial hosts in the natural environment[13,15,17]. We identified potential ecological functions of ARGs in biogeochemical cycling by annotating and mapping all ARG-like reads with genes associated with the cycling of carbon, nitrogen, phosphorus and sulfur (Supplementary Fig. 5). There were 43 genes initially identified as ARGs that clearly perform biological functions in addition to antibiotic resistance. Results indicated that different ARGs exhibit the different level of correlation to the anthropogenic activities, which will influence the health risk of ARGs on human lives. We then quantitatively evaluated the health risk of each ARGs considering the four indicators (human accessibility, mobility, human pathogenicity, and clinical availability) in the following sections.

**Human accessibility of ARGs**. We first examined the ARGs that were shared between humans and the other three main habitats to

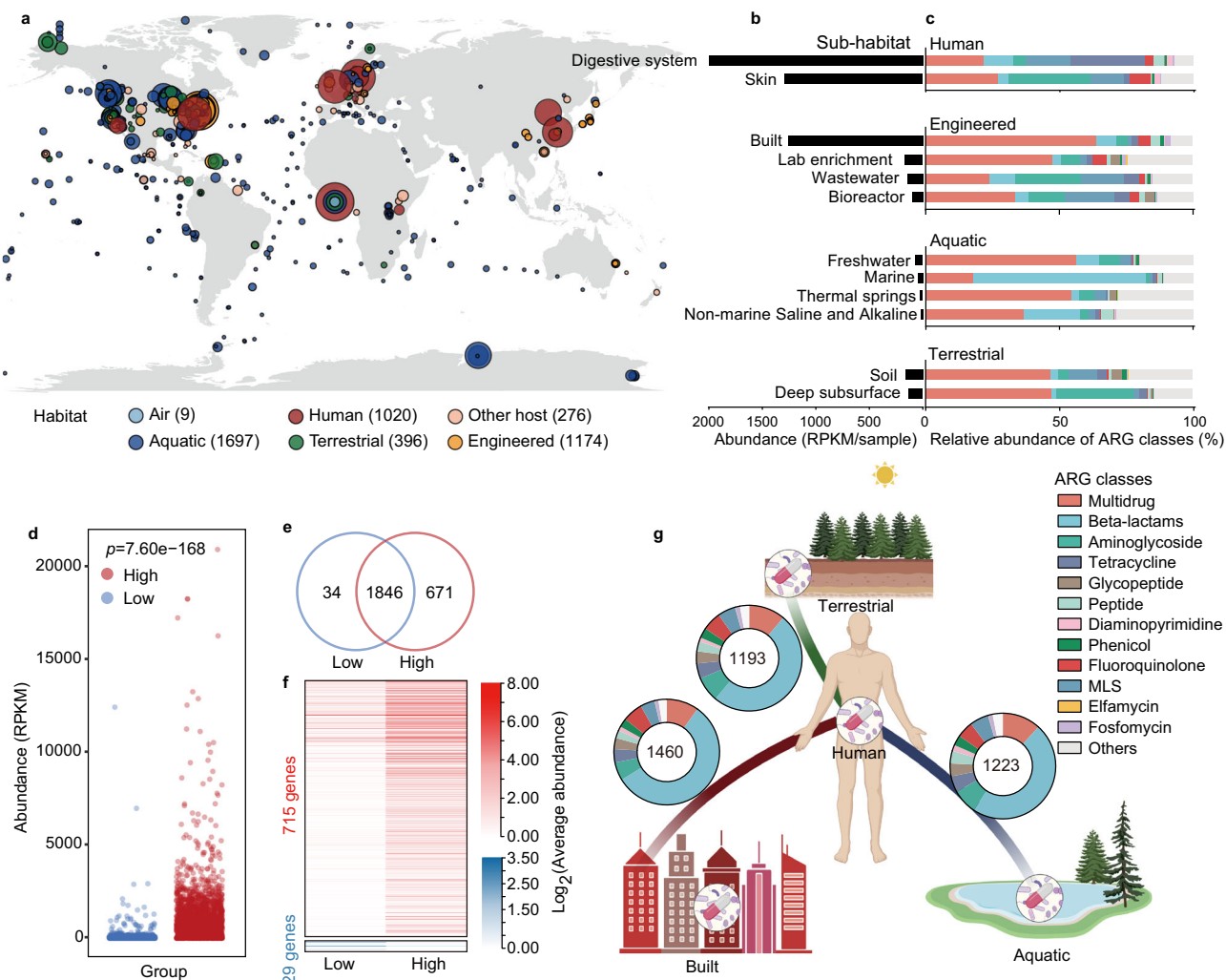

**Fig. 1 Distribution patterns of antibiotic resistance genes (ARGs) globally. a** Geographic distribution of samples with ARG abundance in various habitats. Each point indicates one sampling location, rounded to the nearest degree, with point size reflecting the number of samples, and point color indicating the habitat. Samples with the unclassified location was showed as longitude = 0 and latitude = 0. **b** Abundance of ARGs in each sub-habitat. Digestive system of human, mainly including the fecal samples, had the highest abundances of ARGs. **c** Composition of antibiotic resistome in each sub-habitat. Only sub-habitats containing at least 20 samples are shown. **d** High-intensity human activities significantly promoted the abundance of ARGs. Each dot represents one sample (n = 1643 and 2309 samples for Low- and High-group, respectively). The p value represents the statistical significance (two-tailed Welch's t-test). **e** Number of ARGs specific or shared in the areas with low- or high-intensity human activities. There were 671 ARGs specifically detected in high-intensity human activities environment. **f** The abundance of 715 and 29 ARGs significantly increased in high- and low-intensity human activities environment, respectively (adjust p < 0.05, two-tailed Welch's t-test; Supplementary Data 3). **g** ARGs shared between the human-associated and three main habitats. Number in the circles represents the number of shared ARGs.

investigate the accessibility of ARGs to the human microbiota. As expected, built environments had the most ARGs shared with human habitats (1460), with terrestrial (1193), and aquatic (1223) environments having fewer ARGs (Fig. 1g). Most of these ARGs were annotated as conferring resistance to multidrug and beta-lactams. We then determined the average abundance and prevalence of each ARG in the human habitats and calculated human accessibility (see the "Methods" section; Supplementary Data 4). Only 1714 of 2561 ARGs were detected in the human habitats, most of them with average abundances <50 reads per kilobase per million (RPKM) per sample and prevalence <10% (Supplementary Fig. 6 and Supplementary Data 4). The gene *tet*Q, conferring resistance to tetracycline antibiotics, had the highest human accessibility (Supplementary Fig. 6). These results indicated that the human accessibility of ARGs were variable and only a fraction of ARGs exhibited high accessibility to humans and posed potential risk.

**Distribution of ARG hosts and mobile genetic elements (MGEs) in different habitats.** From the 4572 metagenomic samples we used, 18,465 metagenome-assembled genomes (MAGs) have been recovered by Nayfach et al.[28], which greatly aided our work in determining the global distribution of ARG hosts. However, directly predicting the host of ARGs based on MAGs is challenging and could lead to misleading results because MAGs are composite genomes and do not represent an actual genome of a single microbe in the community[29]. Here, we tried to improve the accuracy of host identification of ARGs by implementing strict quality criteria. We only considered ARGs in contigs longer than 10 kb and made sure that the taxonomic affiliation of any genes found in those ARG-containing contigs agreed with the overall taxonomy of the MAG (Fig. 2a). Subsequently, 7555 MAGs were identified as being host of ARGs (Supplementary Data 5). The hosts of individual ARGs differed significantly in different habitats (Fig. 2b and Supplementary

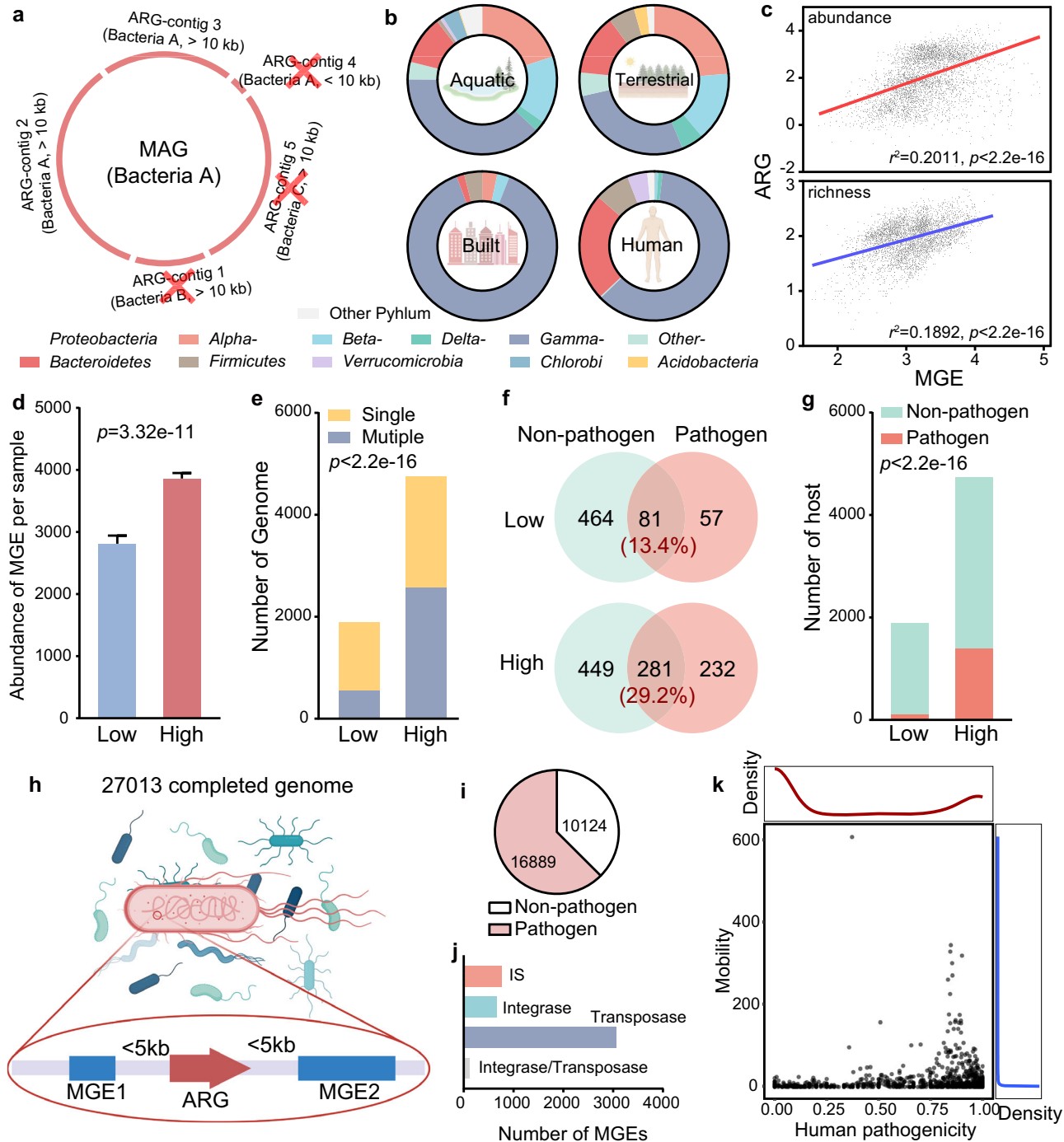

**Fig. 2 Human pathogenicity and mobility of ARGs. a** The host of ARGs were determined by only considering ARGs in contigs longer than 10 kb and making sure that the taxonomic affiliation of those ARG-containing contigs agree with the overall taxonomy of the MAG. **b** Composition of antibiotic resistance host at the phylum level (and classes in *Proteobacteria*) showed the different distributions of ARG hosts in four main habitats. **c** MGEs exhibited a significantly positive correlation with ARGs in abundance and richness. Each point represents one sample ($n = 4572$ samples). The value of abundance and number of ARGs were showed in log10. Result of linear regression are shown as $r^2$ and $p$ value, evaluated by $F$-statistic (one-sided). **d** Abundance of MGEs in high-intensity human activities areas were significantly higher than in low-intensity human activities areas. $n = 1643$ and 2309 biologically independent samples for Low- and High-group, respectively. Significance between two groups was evaluated by two-tailed Welch's $t$-test. Data were showed as mean ± standard error of mean (SEM). **e** Number of genomes containing multiple ARGs increased in high-intensity human activities areas, compared to the low-intensity human activities areas (Fisher's exact test, one-sided). **f** Shared ARGs in pathogens or non-pathogen increased in high-intensity human activities areas, compared to low. The percentage of shared ARGs in the two areas are shown. **g** Pathogenic hosts of ARGs increased in high-intensity human activities areas, compared to low (Fisher's exact test, one-sided). **h** We collected 27,013 completed genome from NCBI RefSeq database for determining the human pathogenicity and mobility of ARGs. Five kb upstream and downstream of the ARGs detected in all completed genome for annotating the MGEs. **i** Among the 27,013 completed genome, 16,889 were recognized as pathogens' genome. **j** We totally identified 4612 MGEs from completed genomes, and most of them were referred to the transposase. **k** The human pathogenicity of ARGs exhibited a bimodal distribution, while most of ARGs (2266/2561) showed mobility <10.

Fig. 7a). Hosts of most ARGs (89.61%) were specific to one habitat, with only few hosts shared across two or three habitats (Supplementary Fig. 7b). ARG hosts in the built and human-associated habitats were less diverse than the hosts in more natural habitats (Fig. 2b), perhaps a consequence of selective pressures from anthropogenic activities.

The habitats contained distinct ARG hosts, suggesting that they directly or indirectly selected these hosts. Mobile genetic elements (MGEs), including insertion sequences (ISs), integrons and transposons, are all capable of horizontally transferring ARGs from one bacterium to another in association with plasmids and phages[21,30]. Abundance and distribution of MGEs can be shaped by environmental selection[31–33]. Thus, we determined the composition of MGEs in each sample from the metagenomic reads to discover if they influence the distribution of ARG hosts in diverse habitats. MGEs were significantly and positively correlated with ARGs in abundance and richness, respectively (linear regression: $p < 0.001$) (Fig. 2c). Few MGEs (3.80%) were shared by the four habitats, similar to the distribution of hosts carried ARG (Supplementary Fig. 8). The unique environmental conditions in each habitat type could influence the distribution of MGEs[31–33] and thus determine the distinct compositions of the hosts that contained ARGs in each habitat. The abundance of MGEs per sample was significantly higher in areas with high-, as opposed to low-intensity anthropogenic activities ($p < 0.001$, two-tailed Welch's $t$-test; Fig. 2d). This could lead to the insertion of more ARGs into genomes (Fig. 2e), which then drives an increase in ARGs shared between non-pathogens and pathogens (Fig. 2f) while also increasing the range of potentially pathogenic hosts of ARGs (Fig. 2g). These results indicate that MGEs likely contributed to the HGT of ARGs between hosts, and significantly from non-pathogens to pathogens.

**Mobility and human pathogenicity of ARGs.** Considering that MGEs can be either not binned, or incorrectly binned into the wrong MAGs[34,35], we collected 27,013 completed genomes in NCBI RefSeq database[36] to determine the mobility and human pathogenicity of ARGs (see the "Methods" section; Fig. 2h). All these completed genomes were sequenced by whole-genome sequencing, the accurate and standard approach for discovering MGEs[35], and 16,889 of them were recognized as pathogens (Fig. 2i and Supplementary Data 6). For determining the mobility of ARGs, we extracted 5 kb upstream and downstream of the ARGs detected in all completed genome for annotating the MGEs (Fig. 2h). We only considered the ISs, integrases and transposases in this step. Some sequences attributed to the function of plasmids and phages, but that did not directly affect the mobility of ARGs were excluded. We used such conservative approach mainly because it is difficult to identify phages and plasmids at the gene level. Some genetic elements close to ARGs may be involved in the function of plasmids and phages, however, they cannot contribute to the HGT of ARGs and result in false positives[21]. In total, there were 4612 MGEs identified from completed genomes, and most of them (3061/4612) were transposase (Fig. 2j).

It is now clearer than ever that MGEs were greatly responsible to the dissemination of ARGs and used for determining the mobility of ARGs in the previous studies, which assessed the health risk of ARGs qualitatively[17,18]. In the present study, for quantitative analysis, the mobility of ARGs was defined as the number of associated MGEs detected (see "Methods" section; Supplementary Data 7). It should be noted that it is almost impossible to measure the absolute value of the mobility of ARG, which can be changed with the genetic contexts in specific species, because of the fitness costs in HGT[17]. However, our method determined a potential mobility of ARGs, which was critical for risk assessment. Most of the evaluated ARGs (2265/2561) were carried by less than 10 different MGEs (Mobility <10) (Fig. 2k).

We also determined the potential human pathogenicity of an ARG based on the proportion of pathogens that carried it to evaluate the health risk of clinical ARGs (see the "Methods" section; Supplementary Data 8). The human pathogenicity of ARGs exhibited a clearly bimodal distribution (Fig. 2k), indicating that most ARGs had specific genetic contexts and were exclusively carried by non-pathogens only (human pathogenicity = 0) or pathogens only (human pathogenicity = 1).

**Health risk assessment of ARGs and samples.** In summary, we analyzed the characteristics of ARGs to determine their human accessibility, mobility, and human pathogenicity based on their potential to move from the environment to humans and to drive the evolution of pathogens resistant to antibiotics. We further determined the clinical relevance of ARGs by systematically evaluating the risk of ARGs to human health. Data for the global use of antibiotics were collected[37], which indicated that penam (55.25%) and cephalosporins (13.07%), two beta-lactam antibiotics, were used the most (Supplementary Fig. 9). The total use of antibiotics for each ARG was calculated as clinical availability (see the "Methods" section and Supplementary Data 9 for details). Genes conferring resistance to clinically available antibiotics were a high proportion of the 2561 ARGs we detected. The multidrug resistance gene *tol*C, for example, can confer resistance to nearly all common antibiotics and thus had the highest clinical availability (Supplementary Data 9).

We evaluated the overall health risk for each ARG using the four calculated metrics: human accessibility, mobility, human pathogenicity and clinical availability, as determined above (Fig. 3a). All, except clinical availability, only covered about half of the ARGs (Fig. 3b). We calculated the risk index (RI) as $RI = HA × MO × HP × CA$ (also see the "Methods" section). This formula was quite strict, and only 23.78% of the 2561 ARGs with all four indicators were predicted as a health risk (Fig. 3c; Supplementary Data 10). Most high-risk ARGs were multidrug resistance determinants (Fig. 3d). However, this formula is still reasonable. For example, the ARGs with high clinical availability but with no MGEs may only be genes intrinsic to specific hosts[17,38] and cannot transfer between hosts. We divided the ARGs with an RI > 0 into four categories based on the rank of their RI for further examination: Q1 (top 25%), Q2 (50–75%), Q3 (25–50%), and Q4 (bottom 25%). Interestingly, most ARGs conferring multidrug resistance or resistance to commonly used antibiotics, such as tetracycline belonged to Q1, and ARGs conferring resistance to rarely used antibiotics such as glycopeptides belonged to Q4 (Fig. 3e). These results confirmed that the use of antibiotics increased the risk of ARGs to human health and potentially caused the failure of clinical treatments of infection[38]. To validate the performance of our assessment method, we determined ARGs in 568 hospital pathogenic MAGs from another catalog of human gut microbiota[39] (Supplementary Data 11) as well as a subset of completed genomes from pathogens. The method used for assigning ARGs in MAGs was the same as Fig. 2a. Results clearly showed that the number of ARGs belonging to Q1 were significantly higher than other ranks per genome (Fig. 3f), confirming that these ARGs were highly dangerous to human health and complicated the clinical treatment of disease. On the contrary, ARGs belonging to Q3 and Q4 were seldomly carried by genomes of pathogens. These results testified to the validity of our work in health risk assessment.

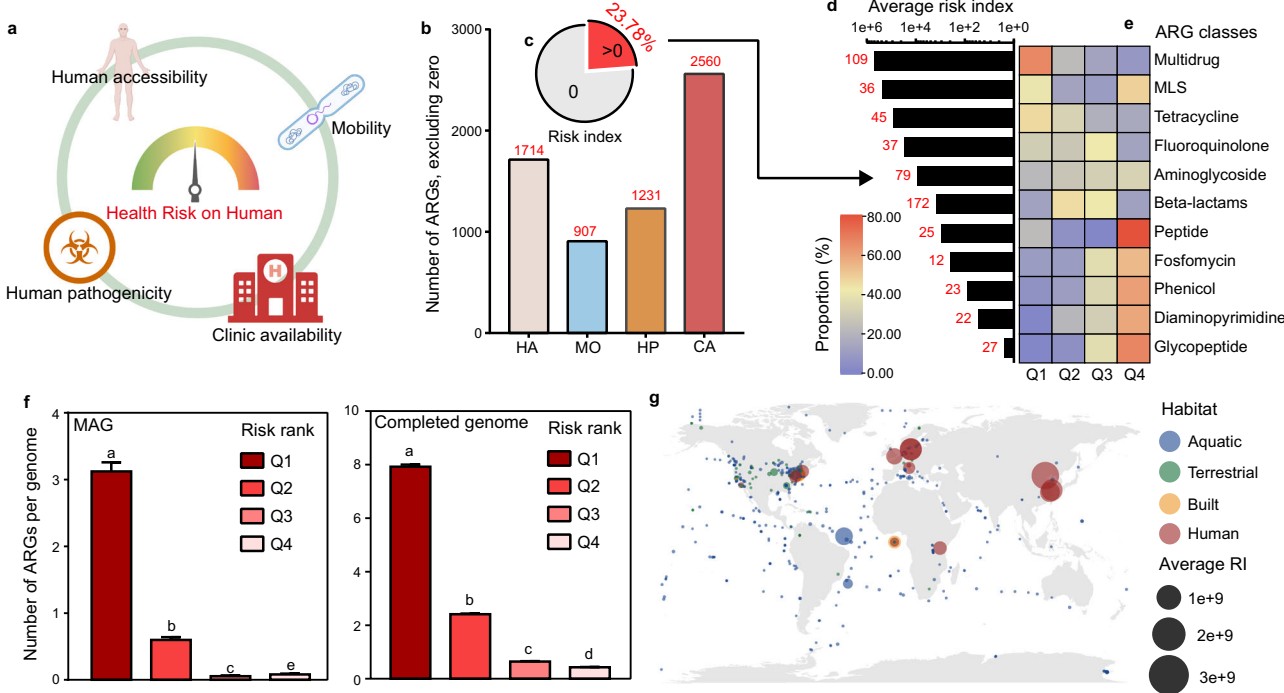

**Fig. 3 Health risk evaluation of ARGs. a** We evaluated the health risk of human for each ARG with four indicators, including human accessibility (HA), mobility (MO), human pathogenicity (HP), and clinical availability (CA). **b** Number of ARGs after excluding the zero value in each indicator for risk index calculation. **c** Only 23.78% of all evaluated ARGs exhibited risk index (RI) > 0. RI of most ARGs was zero because of the strict formula we used. **d** The number and average RI of ARGs in each class. Only classes that had more than 10 ARGs with RI > 0 were shown. Most of the ARGs with RI > 0 were referred to multidrug resistance with the highest average RI. **e** Composition of ARGs with RI > 0 in different classes. Most of the ARGs which resisted commonly used antibiotics belonged to Q1, while most of ARGs which resist rarely used antibiotics belonged to Q4. **f** Number of ARGs per pathogenic genome, which assembled by hospital fecal metagenomes (n = 568 MAGs) or from the completed genome dataset (n = 15,596 MAGs). Pathogens carried much more ARGs belonged to the Q1, compared with other ranks. Different letters represented the significant difference by Kruskal–Wallis *H*-test with the pairwise comparisons. Data were showed as mean ± standard error of mean (SEM). **g** Global ARG risk map of four main habitats. Antibiotic resistance risk was detected all around the world, even in the polar region. Human-associated habitats posed the highest risk of antibiotic resistance than other habitats. The average RI of each sampling site was calculated by the combination of abundance and RI of ARGs and showed as the size of points. Habitats were showed as color. Samples with the unclassified location was showed as longitude = 0 and latitude = 0.

We evaluated the risk of antibiotic resistance for each sample using a combination of abundance and RI of ARGs (see the "Methods" section) to model global surveillance, based on a data set of 4005 metagenomic samples from four main habitats (aquatic, terrestrial, built, and human-associated) around the world. The risk of antibiotic resistance was global, even in the polar region (Fig. 3g). Human-associated habitats posed the highest risk of antibiotic resistance, and health risk was also influenced by anthropogenic activities (Supplementary Fig. S10), as expected.

**Global mapping of the antibiotic resistance threats in marine environments**. After evaluating the health risk of each sample, we wanted to map the antibiotic resistance threats around the world, using machine learning (Fig. 4a). We used 712 samples from marine habitats to establish the prediction model (Supplementary Data 12). The distribution of the risks for marine samples were uneven (Supplementary Fig. 11), so for better prediction accuracy, the dataset was discretized by three unsupervised methods (*k*-means[40], equal width, and equal frequency[41]), and the samples were then divided into 10 ranks according to their risks (rank 10 for the highest risk and rank 1 for the lowest risk). Seventeen anthropogenic drivers in marine habitats provided by Halpern et al.[42] as well as the latitude were chosen as 18 factors (Supplementary Data 12) that influenced the risk ranks. The random forest algorithm combined with 10-fold cross-validation was used

in machine learning for better performance and avoiding the overfitting of prediction model[43].

The different methods discretized the dataset in different results (Supplementary Figs. 12 and 13). Equal frequency resulted in a well-distributed dataset, however, it failed to clearly distinguish the samples in ranks 1–5. On the contrary, equal width clearly differentiated the samples in each rank, but nearly all the samples were grouped as rank 1 with only one sample in some ranks. *k*-means algorithm, the most known and used clustering method[40], balanced the sample number (not strictly even but better than the original dataset and equal width) and dissimilarity in each rank. We further determined the prediction performance of machine learning based on the dataset discretized by *k*-means and equal frequency, and the former exhibited much higher accuracy rate than the latter (Fig. 4b). Results of 10-fold cross-validation also showed the prediction performance of machine learning with the *k*-means method (Supplementary Fig. 14). We then chose the best model in 10-fold cross-validation with the accuracy rate = 76.06% for further analysis. This model classified each risk rank well (Fig. 4c). Latitude, which has been confirmed to significantly influence the abundance of ARGs (Supplementary Fig. 2), was the most important predictor in the prediction model (Fig. 4d). In the meantime, climate change stressors including ultraviolet radiation changes (UV), sea level rise (SLR), surface temperature rise (STR), and ocean acidification (OA) were also critical for the prediction model.

With the prediction results from machine learning, global mapping of antibiotic resistance risk in marine habitats with each

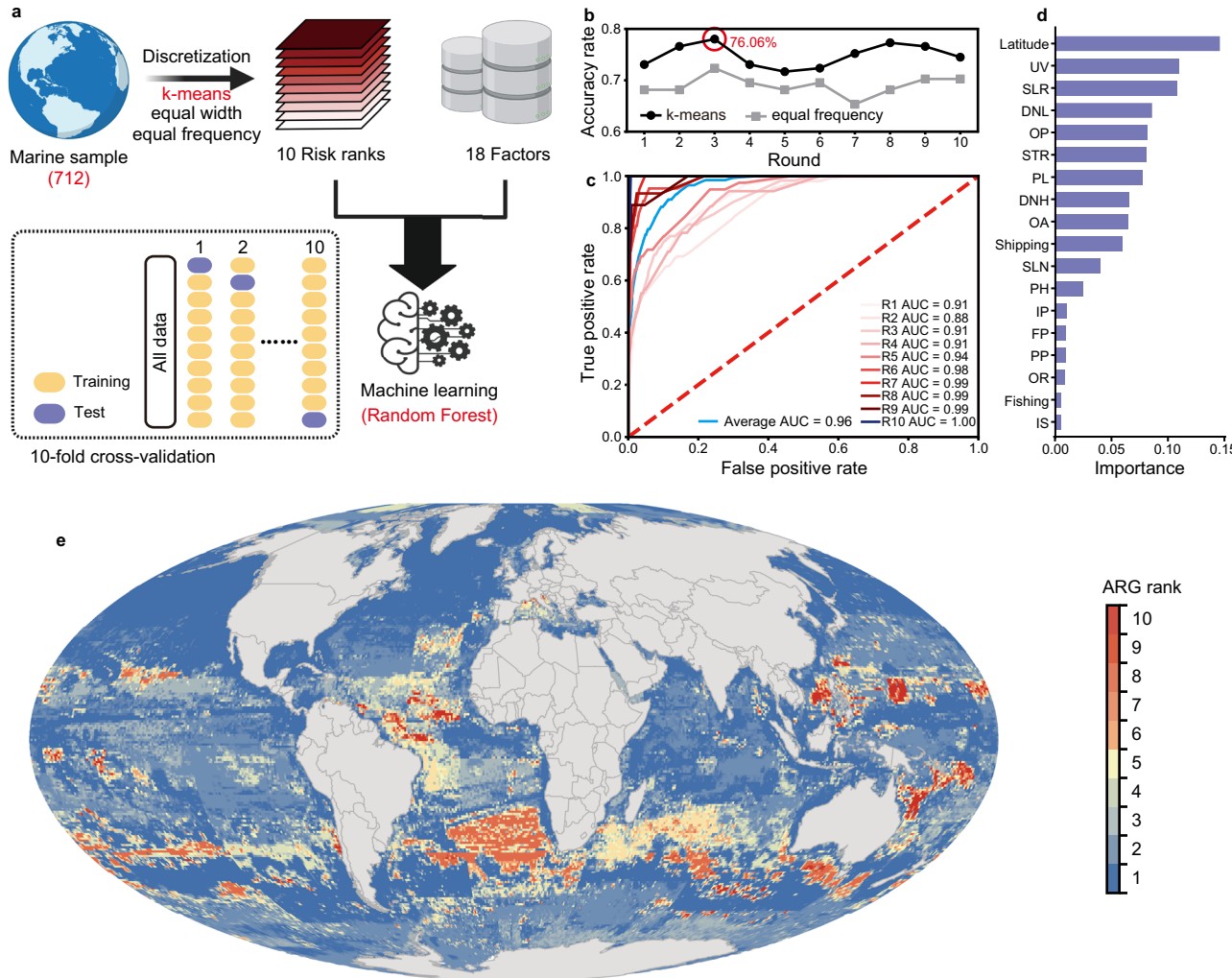

**Fig. 4 Global mapping of antibiotic resistance threats in marine habitats. a** Machine learning were trained by 712 samples from marine habitats and used to predict the antibiotic resistance threats in global marine habitats. **b** Accuracy rate of machine learning with different discretization methods. *K*-means exhibited higher accuracy rate than equal frequency and the best model (accuracy rate = 76.06%) were chosen for the further prediction. **c** The ROC plots confirmed the high performance of the best model in classification of risk ranks. **d** Latitude as well as the climate change stressors exhibited the high importance in predicting the antibiotic resistance risk. The full name of each indicator can be found in Supplementary Data 12. **e** The map of ARG risk in marine habitats with prediction results by machine learning, which was drawn by ArcGIS in 20′ × 20′ resolution.

pixel (20′ × 20′ resolution) were done (Fig. 4e). Note that the risk rank here only represents the relative risk (antibiotic resistance risk in rank 10 areas is higher than rank 1), areas in rank 1 did not mean no risk. The global distribution of antibiotic resistance is related to the prediction of the abundance of ARGs in each country and territory by Hendriksen et al.[5]. For example, the ARG risks in the marine areas close to Brazil and Africa were higher than in the marine areas close to eastern USA. This result was consistent with the higher ARG abundance of urban sewage in Brazil and Africa, compared to the USA[5]. For comparison between oceans, the antibiotic resistance threats in Pacific and Atlantic Ocean were the higher than others. The antibiotic resistance risk of marine areas close to the Antarctic Pole were higher than the areas near the Arctic Pole. This map provided an overview of the antibiotic resistance threat in global marine, which was important for surveillance. However, it is still limited by the number of samples, for example, samples in high-risk ranks were less than lower ones, and there was low coverage of

marine areas, with 712 samples. Thus, for more comprehensive and accurate mapping of antibiotic resistance threats in marine environments, we need more high-quality metagenomic datasets. On the other hand, we could not map the antibiotic resistance threats in terrestrial and human-associated habitats, mainly because of the uneven distribution of samples and the lack of metadata on original soil properties.

## Discussion

We annotated 4572 metagenome samples with CARD to identify ARGs and obtain an unprecedented view of their global ARG distribution. A total of 2561 ARGs covering various habitats were used to link their abundance to the intensity of anthropogenic activities. The dissemination of ARGs across habitats and HGT of ARGs from non-pathogenic hosts to pathogens increased with the intensity of anthropogenic activities, associated with an increase in the abundance of MGEs. These results were predictable, but the global distribution of the main hosts of ARGs

was surprising. By examining different habitats, we demonstrated that particular ARGs were found in distinct host taxa in different environments. Sequencing the 16S rRNA gene combined with high-throughput qPCR is a common approach to identify microorganisms that are ARG hosts by correlation analysis[2,3,44]. Metagenomic sequencing can more accurately identify hosts that have ARGs, although studies are often limited to a single habitat[4,7] and metagenomic approaches can cause false positives. Here we examined all available MAGs from multiple habitats by applying stricter quality criteria, which improved the accuracy of host identification of ARGs.

The global distribution of ARGs and their hosts, mainly based on their number and abundance is important, but insufficient for evaluating the health risk of antibiotic resistance[16–18]. By systematically considering their accessibility, potential to contribute to pathogenicity, mobility between hosts and relevance to clinical treatment, we improved determinations of the health risk for the 2561 ARGs detected in the metagenomic analysis. Only 23.78% of the 2561 ARGs were determined as "risk ARGs" with an RI > 0 because of our stringent, but reasonable, methods. We confirmed that the use of antibiotics was tightly linked to the health risk of the ARGs, but some ARGs were exceptions. For example, ARGs conferring resistance to beta-lactams were common, but only 172 of 1631 ARGs in this class had a significant risk index. ARGs conferring resistance to glycopeptide antibiotics (such as vancomycin, a last line of defense antibiotic) were mainly in the lowest risk Q4 category. For these ARGs, we need information about whether they co-exist with multidrug-resistant ARGs in the same pathogens[30] before the target antibiotics are used. Vaccines would then be more efficient and less risky than antibiotics[45].

Global prediction or surveillance of antibiotic resistance has already begun[5,46,47]. Hendriksen et al. precisely predicted the global abundance of ARGs using World Bank variables, mainly concerning sanitation and health[5], but abundance did not directly represent health risk. In this study, we provided a feasible quantitative method for global mapping of antibiotic resistance threats for each pixel (20′ × 20′) in marine habitats with machine learning. However, there is still an immediate need for a global cooperative system for reporting ARGs and their hosts, especially in clinics. The Global Antimicrobial Resistance Surveillance System of the WHO provides the opportunity for global cooperation on surveilling antimicrobial resistance[47]. Most importantly, these data should be collected under the supervision of governments and shared on public platforms such as the World Health Organization (WHO) with detailed metadata, including clear coordinates of sampling sites, physicochemical properties of samples (for water or soils), age, gender, race and dietary habits (for humans), and so on.

Our study provides a novel and straightforward method for quantification and standardization of the antibiotic resistance threats all around the world that will improve decision-making in clinics and public health management. Predicting future threats, as demonstrated by COVID-19, is difficult. The most serious threats, however, are likely those we already recognize[48]. ARGs and their hosts are unquestionably a serious global threat, currently causing about 700,000 deaths a year around the world, and this could increase to 10 million deaths a year by 2050[48]. We used a systematic and targeted surveillance of ARGs to evaluate the antibiotic resistance threats around the world, which provides some early warning. Comprehensive systems for surveilling the risk of antibiotic resistance under global cooperation should be a priority.

## Methods

**Collecting the datasets of metagenome, completed genomes, and MAGs**. In this study, we used several datasets for comprehensively evaluating the health risk of ARGs. We downloaded 4572 metagenomic samples for determining the global distribution of ARGs and MGEs across diverse habitats from European Nucleotide Archive[49]. We fortunately have the permission to download the 18,465 MAGs constructed by Nayfach et al.[28] from the IMG/M portal[50,51]. For more accurately determining the mobility and human pathogenicity of ARGs, 27,013 completed genomes in NCBI RefSeq database[36] were collected. To validate the performance of our assessment method, we determined ARGs in 568 hospital pathogenic MAGs of another catalog[39] from European Nucleotide Archive[49]. The pathogenic genomes in this study were defined by comparison of their taxonomical information to the A-to-Z database which is continually updated with clinically relevant pathogens and aligned with information from the Robert Koch Institute in Germany and the WHO[52]. Detailed information for these datasets is provided in Supplementary Data 1, 5, 6 and 11, including the ARGs annotation, sampling location, habitat information and pathogenic identification. IBM® Aspera Connect (v4.1.1) was used for downloading all these data.

**ARG and MGE annotation and abundance calculation at the metagenomic level**. The raw data of metagenomic samples were qualified by FastQC (v0.11.5; https://github.com/s-andrews/FastQC), and then trimmed and quality-filtered using Trimmomatic[53] (v0.36). ARGs were annotated with the CARD using reads by their recommended tool, RGI[54] (v5.1.1), with default parameters for metagenomic reads. BWA[55] (v0.7.13) was used for mapping reads to ARGs in each sample, and the unmapped reads were removed using Samtools[56] (v1.3.1). The number of mapped reads of ARGs in each sample were counted using a script, which is available at GitHub (see "Code availability" below). ARG abundance was calculated as RPKM with the number of reads and gene lengths. ARGs were manually reclassified based on the drugs to which they confer resistance. ARGs referring to penam, cephalosporin, carbapenem, cephamycin, penem and monobactam were grouped into the beta-lactam class. ARGs referring to macrolides, lincosamides and streptogramins were grouped into the MLS class. ARGs referring to more than one drug class were grouped into the multidrug class.

The annotation of MGEs in each sample from the metagenomic reads was performed to discover if they could influence the distribution of ARG hosts in diverse habitats. Reads of ISs were annotated with the ISfinder database[57] using BWA[55] (v0.7.13). Reads were also annotated with the nucleotide sequence referred to the NCBI Reference Sequence of transposases and integrases, which were clustered using CD-HIT[58] (v4.7) with the threshold ≥90%, using BWA[55] (v0.7.13). The abundances of the MGEs were calculated in the same manner as for the ARGs.

**Potential ecological functions of ARGs in biogeochemical cycling**. We compiled a comprehensive database of genes involved in the cycling of carbon, nitrogen, phosphorus, and sulfur (CNPS), including SCycDB[59], NCycDB[60], and QMEC[61]. Potential ecological functions of ARG-like reads were annotated with the CNPS database we created by BWA[55] (v0.7.13).

**Determination of the ARG hosts in MAGs**. The MAGs and ARG contigs were taxonomically assigned using Kraken2 (v2.1.2) with the default parameter. After that, we tried to improve the accuracy of host identification of ARGs by implementing stricter quality criteria: only considering ARGs in contigs longer than 10 kb and making sure that the taxonomic affiliation of any genes found in those ARG contigs agreed with the overall taxonomy of the MAG (Fig. 2a).

**ARGs and MGEs annotation in completed genomes**. ARGs in completed genomes were annotated with CARD by their recommended tool, RGI[54] (v5.1.1), with default parameters for genomes. We then extracted 5 kb upstream and downstream of the ARGs detected in all completed genomes for annotating the MGEs, because such close proximity of MGEs and ARGs is more likely to induce HGT[21]. For genomes, ISs were annotated with the ISfinder database[57] using BLASTN (v0.7.13; $e$-value ≤ $10^{-10}$, identity ≥ 80%, coverage ≥ 80%), while the transposases and integrases were annotated with the NCBI Reference Sequence, which were clustered using CD-HIT[58] (v4.7) with the threshold ≥ 90%, at an $e$-value ≤ $10^{-10}$ with a minimum amino acid identity of 60% over 60% query coverage using Diamond[62] (v0.9.36.137).

**Collection of population data and global antibiotic use**. The population density of each sampling location was collected from the SEDAC Population Estimation Service[63], which provides population data in a specific area using longitude and latitude. We only considered samples from aquatic, terrestrial, engineered and human-associated habitats with clear location coordinates. All samples were clustered by population density into two groups regardless of habitat: one was an area with high-intensity anthropogenic activities with >58 people/km², and the other area was summarized as low-intensity activity. This threshold was chosen because it is the global average population density calculated according to the global population and land area[27]. Data for global antibiotic use were collected from an online website[37] containing data on antibiotic use in hospitals and from retail outlets in 76 countries.

**Determining the health risk of ARGs to humans**. We emphasized the importance of determining the health risk of ARGs to humans from multiple perspectives rather than basing risk only on abundance. We defined four indicators from these perspectives: "human accessibility" (HA), "mobility" (MO), "human pathogenicity" (HP), and "clinical availability" (CA).

HA represents the ability of ARGs to transfer from the environment to the bacterial groups in humans[16] and is calculated as

$$HA = Average\ abundance_{human} \times Prevalence_{human} \quad (1)$$

where Average abundance$_{human}$ and Prevalence$_{human}$ represent the average abundance and prevalence of ARGs in human-associated habitats, respectively.

MO represents the ability of ARGs to transfer between hosts by HGT. We determined the MO of ARGs as the number of the related MGEs in completed genomes, which could potentially transfer them between genomes or to plasmids.

HP specifically represents the ability of ARGs to transfer from nonpathogenic hosts to pathogenic hosts, leading to the evolution of pathogens resistant to antibiotics and the failure to control clinical infections[17]. HP was calculated for each ARG as

$$HP = Number_{pathogenic} / Number_{all} \quad (2)$$

where Number$_{pathogenic}$ is the number of pathogenic ARG hosts and Number$_{all}$ is the total number of hosts containing ARGs.

CA represents the clinical availability of ARGs, so we indicated that their health risk was higher than others if the ARGs conferred resistance to the most commonly used antibiotics. We then calculated CA for each ARG as

$$CA = \sum_{i=1}^{n} Usage\ of\ antibiotic_i \quad (3)$$

where $n$ is the number of antibiotics to which the ARG conferred resistance. For example, according to the classification in CARD, *tetC* confers resistance to macrolide, fluoroquinolone, aminoglycoside, carbapenem, cephalosporin (cephamycin), glycylcycline, penam, tetracycline, peptide, aminocoumarin, rifamycin, phenicol, triclosan, and penem antibiotics. So, the number of antibiotics to which *tetC* conferred resistance is 14, and the clinical availability of *tetC* was the sum of the consumption of these 14 antibiotics.

Finally, the risk index (RI) of the ARGs to human health was calculated as

$$RI = HA \times MO \times HP \times CA \quad (4)$$

We also calculated RI for each sample as

$$RI_{sample} = \sum_{i=1}^{n} Abundance_i \times RI_i \quad (5)$$

where Abundance$_i$ is the abundance of ARG$_i$ in the sample, and RI$_i$ is the RI of ARG$_i$ calculated using Eq. (4).

To validate the performance of our assessment method, the ARG annotation from hospital pathogenic MAGs were performed as described above. We calculated the number of ARGs carried by per pathogenic MAG with different risk ranks (Q1–Q4), respectively. We also compared the number of ARGs in each risk rank detected in the pathogenic completed genomes.

**Global mapping for antibiotic resistance threats in marine environments**.
Seventeen data sets of anthropogenic drivers in marine habitats were collected from the database provided by Halpern et al.[42] and downloaded from https://knb.ecoinformatics.org/view/doi:10.5063/F1S180FS in August 2021 and used for prediction the antibiotic resistance threats in marine habitats. In the meantime, latitude was chosen as a geographic factor because it has been reported to influence the abundance of the ARGs[4], as confirmed in this study. The 712 samples from marine habitats were divided into 10 ranks according to their risk after discretization. The 18 selected indicators associated with the 712 marine samples were collected for machine learning using random forest algorithm combined with the ten-fold cross-validation. Random forest was an accurate algorithm by using bootstrap sample: each tree was built by about 2/3 samples of all data and model performance was validated by the remaining out-of-bag data[64]. To further ensure the performance of random forest and avoid overfitting, we used 10-fold cross-validation. The original dataset was randomly partitioned into 10 folds. In each round, nine folds were used to train the model as the training set, and the remaining data evaluated model as the test set[43]. The results of ten-fold cross-validation were evaluated by confusion matrix. The final prediction model was constructed based on the $k$-means discretization methods because it exhibited the best performance. We further evaluated the performance of machine learning by receiver operator characteristic curve (ROC) plots, which constructed in each rank. The importance of each indicator was also determined by machine learning to estimate the most critical factors influencing the antibiotic resistance threats in marine habitats. We also collected the information of 18 indicators in 380,887 sites with $20' \times 20'$ resolution in marine areas for prediction. After calculating risk for each site, we converted all data to the Mollweide projection with a WGS84 datum for showing the global antibiotic resistance threats in marine habitats by ArcGIS (v10.8). All the scripts for machine learning, including building the prediction model, plotting ROC, determining the importance of indicators, and predicting the risk of unobserved marine sites, can be found in the GitHub (see the "Code availability" section). Machine learning were performed in Python (v3.9) using PyCharm Community Edition (version: 2021.2.2).

**Statistical analysis and visualization**. Significant differences were identified using multiple methods, mentioned in the main text and legends of figures. Two-tailed Welch's $t$-tests were performed in Excel Analysis Tools (Microsoft Corporation, Redmond, USA), $F$-statistic and Fisher's exact test were performed by the basic statistic package in R (v3.6.3), and Kruskal–Wallis $H$-test with the pairwise comparisons were performed in IBM® SPSS® Statistics (v20.0.0). For multiple comparisons, $p$ values were adjusted by FDR in R (v3.6.3). The geographic distribution of the samples and the global RI map of diverse habitats was constructed using ggmap[65] and ggplot2[66] packages in R (v3.6.3). All Venn diagrams were generated using EVenn (http://www.ehbio.com/test/venn; v1.0). The regression analysis, point plots, density plots and box plots visualized using the ggplot2 package in R (v3.6.3). All the schematic diagrams or elements in this paper were drawn using BioRender (https://app.biorender.com/; v1.0) with full publishing rights. Heatmaps were constructed using TBtools[67] (v1.082). Other plots were constructed using GraphPad Prism (v7.00).

**Reporting summary**. Further information on research design is available in the Nature Research Reporting Summary linked to this article.

## Data availability
All raw data used in this study are available in NCBI RefSeq database, IMG/M portal, and European Nucleotide Archive. Information for all metadata used in this study as well as the important data for analysis are provided in Supplementary Data 1, 5, 6 and 11, including the accession-codes. Supplementary Data contained the critical supplementary information in this study are also publicly available online (https://doi.org/10.6084/m9.figshare.19189652). The raw data underlying figures are provided as Source data which can be obtained in public repository (https://doi.org/10.6084/m9.figshare.19189568). The nucleotide sequences in our CNPS database can be available in https://doi.org/10.6084/m9.figshare.19193489. Source data are provided with this paper.

## Code availability
All the scripts and codes for gene annotation, abundance calculation, machine learning, statistical analysis and visualization used in this study are available online at https://github.com/ZhenyanZhang/ARG-global[68].

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

## Acknowledgements

We appreciate Nayfach et al. (2020)[28] for their work on the genomic catalog of Earth's microbiomes and its publication as a public resource, and Almeida (2021)[40] for their work on the human gut genome and its publication as a public resource. We also acknowledge all the PIs for uploading the metagenomic samples as precious public resources. All the PIs' name were listed in Supplementary Data 1. H.F.Q. was financially supported by the National Natural Science Foundation of China (21976161 and 21777144). T.L. was financially supported by the National Natural Science Foundation of China (41907210). Josep Penuelas was financially supported by the Fundación Ramon Areces grant ELEMENTAL-CLIMATE, the Spanish Government grant PID2019-110521GB-I00, and the Catalan Government grants AGAUR-2020PANDE00117 and SGR 2017-1005.

## Author contributions

Z.Y.Z. and Q.Z. designed the study with guidance from H.F.Q. Z.Y.Z. wrote the first draft of the manuscript and H.F.Q., J.P., and M.G. contributed substantially to revisions. W.J.H., M.X.W., and W.W.G. contributed to the all functional annotation of metagenome-assemble genomes. T.Z.W., Z.Y.Z., and Q.Z. performed all the metagenomic analysis. N.H.X. and T.L. were responsible for the Machine learning model construction and related data analysis. Z.Y.Z. and Q.Z. performed the visualization of all data and the artistic design of all figures. H.F.Q., T.L., and J.P. acquired funding for this project.

## Competing interests

The authors declare no competing interests.
