## [Peer Review File · Nature Communications]

Reviewer comments, initial review –

Reviewer #1 (Remarks to the Author):

I carefully read the manuscript "Assessment of global health risk of antibiotic resistance genes". This paper proposed a new quantitative method for global mapping of antibiotic resistance threats on human health by annotating the MAGs with ARGs and MGEs. This is a well-written paper with some noteworthy results, however, there are some minor problems that need to be addressed:

1. In line 125, the authors divide the samples into two groups bounded by 10 people / km². The reason or reference of this classification standard should be given in the manuscript.
2. In Figure 4 and the "Methods" section (line 650), the authors categorized the data into 11 ranks considering the average risk, but how are these 11 ranks categorized, and what are their minimum, maximum and interval value?
3. Some citations lack information such as page numbers (e.g. Ref. 11, 12).
4. Figure 4 is a significant research result, but it is difficult to fully display it in the form of a picture. It would be better to add the specific data or interactive diagrams to the supplementary files.

Reviewer #2 (Remarks to the Author):

Zhang et al. provide a comprehensive characterization of the composition and diversity of Antibiotic Resistance Genes (ARGs) in a global collection of Metagenome-Assembled Genomes (MAGs) from Nayfach et al. 2021. They find a large collection of ARGs and link their distribution with the sample origin, host and anthropogenic activities.

This type of work is important to start taking advantage of the large collections of genomes that are becoming increasingly available for making new biological discoveries. There are also some interesting analyses that the authors carried out to make inferences about the risk of antibiotic resistance exposure and emergence throughout the world.

However, underpinning the whole manuscript is a fundamental limitation that raises serious doubts about any conclusions described in their work: MAGs cannot be reliably used for making any inferences about horizontal gene transfer, mobile genetic elements and the genes they harbour. Metagenome-assembled genomes represent population genomes (i.e., species-specific) that are reconstructed by exploiting sequence similarities in terms of coverage and composition. As MGEs are, by definition, only found in some strains and not in others, their coverage profiles will be significantly different from the rest of the core genome. As recently described (see: PMID 33001022), this means that all types of MGEs (genomic islands, plasmids, phages etc.) are often either not binned, or incorrectly binned into the wrong MAGs. As most ARGs are known to be harboured by MGEs, this causes several major issues.

Therefore, my main suggestion is for the authors to entirely reframe their study by investigating ARGs at a metagenomic read level, instead of at the MAG level. Identifying the ARG hosts would still be challenging, but I would be much more confident about the abundance, prevalence and composition patterns derived from the metagenomic reads that contain all the sequenced data. I provide below more specific comments regarding my main concerns:

1) Lines 104-105. The fact that only 1% of the MAGs contained any ARGs was a red flag to me. Given the aforementioned issues on binning MGEs, there is likely to be a huge number of false negatives (i.e., MAGs where no ARG was detected but whose genomes they represent actually do carry ARGs). What is the estimated proportion of genomes in nature that carry at least one ARG? My assumption is that this is an order of magnitude higher.

2) Line 158. Along the same line, predicting the host of the ARG based on the MAG they belong can be very challenging and lead to misleading results. The authors need to implement stricter quality criteria to assign a host. This could be for instance by only considering ARGs in contigs longer than a certain threshold (e.g. 10 kb) and by making sure that the taxonomic affiliation of any genes found in those ARG contigs agree with the overall taxonomy of the MAG. If some of those MGEs are found in public databases (e.g. NCBI nr), it would also be useful to determine whether the same MGE-host association is found in isolate genomes.

3) Given that the success of recovering MGEs in a MAG is also dependent on the level of sequencing depth and sample (i.e., strain) complexity, any patterns of ARG abundance and prevalence inferred from MAGs obtained from different samples are likely not biologically meaningful.

4) Line 111. I have some major concerns regarding the way the authors are calculating and interpreting the abundance of the ARGs. The authors calculated the abundance of ARGs based on mapping the reads to the MAGs carrying those ARGs. The problem here is that it is not possible to distinguish the abundance of the MAG from the abundance of the ARG. If one MAG carrying one ARG is highly abundant because it is well adapted to a particular habitat, any ARGs present in this MAG will also be detected as highly abundant, but for reasons that could be entirely unrelated with the ARG itself. It would be more meaningful for the authors to calculate the overall abundance of ARGs at the metagenomic level using a more comprehensive database of ARGs (i.e., how many metagenomic reads are mapped to any ARGs, irrespective of which were detected in MAGs). Tools like GROOT can be used for this purpose: <https://github.com/will-rowe/groot>

5) Lines 179-180. I do not understand the reasoning behind this. It is well established that plasmids act as vectors of ARG transmission, so saying that they “do not directly affect the mobility of ARGs” is very strange to me. I think the main issue here is that due to the inherent properties of MAGs, analysing plasmids and phages for ARG mobility using MAGs alone is almost impossible.

6) Line 85: “ARGs integrated into the chromosome of MAGs are more stable and heritable” -> “ARGs integrated into the chromosome are more stable and heritable” as this is true across all genomes, not just MAGs.

Reviewer #3 (Remarks to the Author):

Zhang et al conducted antibiotic resistance gene profiling through earth microbiome sequences and correlated the results with human habits features. Their main conclusion is that ARG poses health risk and a prediction model is built for global marine. This work is novel and can be significant. It builds on the existing large-scale microbiome data collected earthwide.

The analysis and methodology have ambiguities. Revision is needed and I have provided detailed

comments for the authors at the end.

I examined the GitHub repository provided by the authors. Due to its brevity, reproducing the key results are not possible yet.

Detailed comments:

1. Line 31, "... largely driven by the host range". Please explain the meaning of "host range".
2. Line 34, "Our results demonstrated that about 25% of the ARGs posed a health risk," based on the analysis, is the 25% arbitrarily determined by the author (see Line 227)?
3. Line 37, "...global health risk...". Please explain more on this term "global health risk" in the abstract.
4. Line 81, "... it is the first analysis of ARGs from MAGs rather than from unassembled sequences." I searched "assembled metagenome genomes antibiotic resistance", and found this manuscript analyzes ARGs detected from MAG (10.1021/acs.est.5b03522) in 2019.
5. Line 114. "Genes conferring multidrug resistance dominated in most sub-habitats, especially in human, engineered, freshwater and terrestrial environments (Fig. 1c). These results implied that, in addition to geographic factors (Fig. S2), anthropogenic use of antibiotics influenced the composition and abundance of the ARGs." The logic implication is not clear: as all environments have their specific geographic factors that can influence ARGs, based on the figures, it is not convincing that anthropogenic use of antibiotics has the effects to influence ARGs as well.
6. Line 140, in figure S5, only a few genes were analyzed. But at the beginning of this paragraph "One hundred and ninety-one ARGs...". The author needs to explain the discrepancies in the number of genes.
7. Line 141, "These genes, initially identified as ARGs, clearly perform biological functions other than antibiotic resistance¹⁹. I am not clear about the conclusion. It seems these ARGs can perform AR related function according to RGI annotation. I do not understand why they do not in this sentence.
8. Line 156, "This result confirms the influence of antibiotic use on the distribution of ARGs, because these are commonly used antibiotic classes." Can authors explain whether *adeF* has some relationship to fluoroquinolone and tetracycline? Since *adeF* is used as an example here, it is necessary to examine or explain its relationship to antibiotics.
9. Line 184, "ARGs could be linked to more than one MGE (Fig. S8)." Please explain how to determine the "link" between ARG and MGE.
10. Line 203, "...the hosts of 125 ARGs were all pathogens (human pathogenicity = 1; Table S2)." I cannot find the method descriptions on how to determine the hosts of ARGs. Table S2 does not list host either.
11. Line 264, when building the machine learning model, what is the sample sizes for the training

and test datasets? What is the most important predictor in the prediction model? Are the responses categorical variables (11 categories)? If so, how many samples are in each category? Can you provide ROC plots?

12. Figure 2a, please explain the meaning of the circles and zoom circles.

13. Line 580, "...by Eloe-Fadrosh et al (2020)²⁸". The citation style is not correct, as here refers to the name of the last author, but the name of the first author should be used.

14. Line 612, "...the number of antibiotics to which the ARG conferred resistance." Please explain how the number is calculated.

15. Definition of pathogenic host is not clear. By looking at the A-to-Z database (<https://www.bode-science-center.com/center/relevant-pathogens-from-a-z.html>), I am not sure how to use it to determine pathogenic and non-pathogenic host.

A relevant question is about "We also determined the potential human pathogenicity of ARGs based on the proportion of pathogens in their hosts to evaluate the health risk of clinical ARGs.". Here, how do you determine the proportion of pathogens? Are "hosts" referring to human in this sentence?

REVIEWER COMMENTS

Reviewer #1 (Remarks to the Author):

I carefully read the manuscript “Assessment of global health risk of antibiotic resistance genes”. This paper proposed a new quantitative method for global mapping of antibiotic resistance threats on human health by annotating the MAGs with ARGs and MGEs. This is a well-written paper with some noteworthy results, however, there are some minor problems that need to be addressed:

Response: Thank you so much for your positive assessment and insightful comments, which have greatly improved our work. We hope our study can draw more attention on quantifying and surveilling the health risks of the widespread antibiotic resistance.

1. In line 125, the authors divide the samples into two groups bounded by 10 people / km². The reason or reference of this classification standard should be given in the manuscript.

Response: Sorry for the unclear description on the methods. Ten people / km² were chosen because we considered it was low enough to define as “low population density”, however, there was no reference for this threshold. So, we changed the threshold to the global average population density¹ (58 people/km²) in the revised manuscript (line 115) and re-analyzed all the related data. We think it is more reasonable and we thank again your valuable comment.

2. In Figure 4 and the "Methods" section (line 650), the authors categorized the data

into 11 ranks considering the average risk, but how are these 11 ranks categorized, and what are their minimum, maximum and interval value?

Response: Sorry for not providing such important information. In the revised manuscript, we have added more details about machine learning and changed the 11 ranks to 10 ranks with the new data, because the abundance of ARGs were re-calculated on the metagenomic reads level as suggested by reviewer #2 and thus all the related data were re-analyzed. We used 712 samples from marine habitats to establish the prediction model (Supplementary Table 11). The distribution of the risks for marine samples were uneven (Supplementary Fig. 11), so for better prediction accuracy, the dataset was discretized by three unsupervised methods (k-means, equal width, and equal frequency)^{2,3}, and the samples were then divided into 10 ranks according to their risks (rank 10 for the highest risk and rank 1 for the lowest risk). The different methods discretized the dataset in different results (Supplementary Figs. 12, 13). Equal frequency resulted in the well-distributed dataset, however, it failed to clearly distinguish the samples in ranks 1 to 5. On the contrary, equal width clearly differentiated the samples in each rank, but nearly all the samples were grouped as rank 1 and only one samples in some ranks. K-means algorithm, the most known and used clustering method², balanced the sample number (not strictly even but better than the original dataset and equal width) and dissimilarity in each rank. The number of samples as well as the minimum, maximum and interval value for each rank is shown in Supplementary Fig. 12 and 13. More details can be found in the manuscript (lines 262-269, 274-280), and we hope we have described it clearly now.

3. Some citations lack information such as page numbers (e.g. Ref. 11, 12).

Response: Sorry for the mistake. All the references have been checked in the revised manuscript.

4. Figure 4 is a significant research result, but it is difficult to fully display it in the form of a picture. It would be better to add the specific data or interactive diagrams to the supplementary files.

Response: Thank you again for your insightful suggestions. Figure 4 as well as the “Global mapping of the antibiotic resistance threats in marine habitats” section were re-analyzed and improved as follows: 1) we have added more details in the methods of machine learning and data discretization; 2) we have comprehensively determined the efficiency and accuracy of machine learning; 3) we have estimated the importance of each indicator in machine learning; 4) we have re-mapped the global antibiotic resistance threats in marine habitats using new data. We have also uploaded all the data related to Figure 4 in Supplementary Data 1.

Reviewer #2 (Remarks to the Author):

Zhang et al. provide a comprehensive characterization of the composition and diversity of Antibiotic Resistance Genes (ARGs) in a global collection of Metagenome-Assembled Genomes (MAGs) from Nayfach et al. 2021. They find a large collection of ARGs and link their distribution with the sample origin, host and

anthropogenic activities.

This type of work is important to start taking advantage of the large collections of genomes that are becoming increasingly available for making new biological discoveries. There are also some interesting analyses that the authors carried out to make inferences about the risk of antibiotic resistance exposure and emergence throughout the world.

However, underpinning the whole manuscript is a fundamental limitation that raises serious doubts about any conclusions described in their work: MAGs cannot be reliably used for making any inferences about horizontal gene transfer, mobile genetic elements and the genes they harbour. Metagenome-assembled genomes represent population genomes (i.e., species-specific) that are reconstructed by exploiting sequence similarities in terms of coverage and composition. As MGEs are, by definition, only found in some strains and not in others, their coverage profiles will be significantly different from the rest of the core genome. As recently described (see: PMID 33001022), this means that all types of MGEs (genomic islands, plasmids, phages etc.) are often either not binned, or incorrectly binned into the wrong MAGs. As most ARGs are known to be harboured by MGEs, this causes several major issues. Therefore, my main suggestion is for the authors to entirely reframe their study by investigating ARGs at a metagenomic read level, instead of at the MAG level. Identifying the ARG hosts would still be challenging, but I would be much more confident about the abundance, prevalence and composition patterns derived from the metagenomic reads that contain all the sequenced data.

Response: Thank you so much for your positive evaluation, comments and valuable suggestions. Your comments greatly improved our work and made our story much sounder and clearer. We have re-analyzed all data and rewritten the manuscript following your suggestions. The global abundance, prevalence and composition patterns of ARGs and MGEs in diverse habitats were now determined and discussed on the metagenomic reads level that contain all the sequenced data rather than MAGs. For the distribution of ARG hosts, we identified these hosts by implementing stricter quality criteria also following the reviewer's insightful suggestion: only considering ARGs in contigs longer than 10 kb and making sure that the taxonomic affiliation of any genes found in those ARG contigs agree with the overall taxonomy of the MAG (Fig. 2a). For the determination of the mobility and human pathogenicity of ARGs, we collected 27013 completed genomes in NCBI RefSeq database⁴ (see "Methods" section; Fig. 8h). All these completed genomes were sequenced by whole-genome sequencing, the accurate and standard approach for discovering the MGEs⁵. We agree with the referee that the re-analysis with such workflows is sounder and more accurate for quantifying the risk of ARGs. Thanks for making us notice it.

I provide below more specific comments regarding my main concerns:

1) Lines 104-105. The fact that only 1% of the MAGs contained any ARGs was a red flag to me. Given the aforementioned issues on binning MGEs, there is likely to be a huge number of false negatives (i.e., MAGs where no ARG was detected but whose genomes they represent actually do carry ARGs). What is the estimated proportion of genomes in nature that carry at least one ARG? My assumption is that this is an order

of magnitude higher.

Response: Sorry for the unclear description which misled the reviewer. When stating that “Most of ARGs were detected from less than 1% of the MAGs” in the original manuscript we aimed to mean that the frequency of most of ARGs is less than 1% rather than “only 1% of the MAGs contained any ARGs”. In the revised manuscript, we revised this description as “the frequency of most ARGs (2313/2561) were less than 10%” (line 95). It’s challenging to determine how many genomes in nature carry at least one ARG by using the dataset in this study, but we do agree that this is an order of magnitude higher than 1%, in agreement with the reviewer’s assumption.

2) Line 158. Along the same line, predicting the host of the ARG based on the MAG they belong can be very challenging and lead to misleading results. The authors need to implement stricter quality criteria to assign a host. This could be for instance by only considering ARGs in contigs longer than a certain threshold (e.g. 10 kb) and by making sure that the taxonomic affiliation of any genes found in those ARG contigs agree with the overall taxonomy of the MAG. If some of those MGEs are found in public databases (e.g. NCBI nr), it would also be useful to determine whether the same MGE-host association is found in isolate genomes.

Response: We thank the reviewer for his/her insightful suggestion, which made our analysis much more sound. For the distribution of ARG hosts, we identified these hosts by implementing stricter quality criteria following the reviewer’s insightful suggestion: only considering ARGs in contigs longer than 10 kb and making sure that

the taxonomic affiliation of any genes found in those ARG contigs agree with the overall taxonomy of the MAG (Fig. 2a). For the determination of the mobility and human pathogenicity of ARGs, we collected 27013 completed genomes in NCBI RefSeq database⁴ (see “Methods” section; Fig. 8h). We think these workflows and dataset can be more credible for linking the hosts and MGEs to ARGs.

3) Given that the success of recovering MGEs in a MAG is also dependent on the level of sequencing depth and sample (i.e., strain) complexity, any patterns of ARG abundance and prevalence inferred from MAGs obtained from different samples are likely not biologically meaningful.

Response: We thank the reviewer for pointing this out. In the revised manuscript, the global abundance, prevalence and composition patterns of ARGs in diverse habitats were determined and discussed on the metagenomic reads level that contain all the sequenced data rather than MAGs.

4) Line 111. I have some major concerns regarding the way the authors are calculating and interpreting the abundance of the ARGs. The authors calculated the abundance of ARGs based on mapping the reads to the MAGs carrying those ARGs. The problem here is that it is not possible to distinguish the abundance of the MAG from the abundance of the ARG. If one MAG carrying one ARG is highly abundant because it is well adapted to a particular habitat, any ARGs present in this MAG will also be detected as highly abundant, but for reasons that could be entirely unrelated with the

ARG itself. It would be more meaningful for the authors to calculate the overall abundance of ARGs at the metagenomic level using a more comprehensive database of ARGs (i.e., how many metagenomic reads are mapped to any ARGs, irrespective of which were detected in MAGs). Tools like GROOT can be used for this purpose: <https://github.com/will-rowe/groot>

Response: We fully agree with the reviewer's comments, so we have re-analyzed the data accordingly in the revised manuscript. The global abundance, prevalence and composition patterns of ARGs in diverse habitats were now determined and discussed on the metagenomic reads level that contain all the sequenced data rather than MAGs. ARGs were annotated with the CARD using reads by their recommended tool, RGI⁶ (v5.1.1), with default parameters for metagenomic reads. BWA⁷ (v0.7.13) was used for mapping reads to ARGs in each sample rather than MAG, and the unmapped reads were removed using Samtools⁸ (v1.3.1). The number of mapped reads of ARGs in each sample were counted using a script). ARG abundance was then calculated as RPKM with the number of reads and gene lengths. All the codes and scripts used in this study are available at GitHub (<https://github.com/ZhenyanZhang/ARG-global>).

5) Lines 179-180. I do not understand the reasoning behind this. It is well established that plasmids act as vectors of ARG transmission, so saying that they “do not directly affect the mobility of ARGs” is very strange to me. I think the main issue here is that due to the inherent properties of MAGs, analysing plasmids and phages for ARG mobility using MAGs alone is almost impossible.

Response: We agree with the reviewer that determining plasmids and phages at the genus level is difficult. Some genetic elements closed to ARGs may be involved in the function of plasmids and phages, however, they cannot contribute in the HGT of ARGs and result on the false positives⁹. Thus, we only considered the ISs, integrase and transposase in this step and some sequences attributed to the function of plasmids and phages but may not directly affect the mobility of ARGs were excluded. We included this explanation in the revised manuscript (line 198-204).

6) Line 85: “ARGs integrated into the chromosome of MAGs are more stable and heritable” -> “ARGs integrated into the chromosome are more stable and heritable” as this is true across all genomes, not just MAGs.

Response: We agree with the reviewer’s suggestion. Furthermore, we have now deleted this description in the revised manuscript because we think it does not correspond with the re-analysis on metagenomic reads level.

Reviewer #3 (Remarks to the Author):

Zhang et al conducted antibiotic resistance gene profiling through earth microbiome sequences and correlated the results with human habits features. Their main conclusion is that ARG poses health risk and a prediction model is built for global marine. This work is novel and can be significant. It builds on the existing large-scale microbiome data collected earthwide.

Response: Thank you so much for your positive assessment of our research you're your insightful comments, which have greatly improved our work.

The analysis and methodology have ambiguities. Revision is needed and I have provided detailed comments for the authors at the end.

Response: Sorry for the unclear descriptions on analysis and methodology. We carefully re-analyzed and revised the whole manuscript following your suggestions as detailed below.

I examined the GitHub repository provided by the authors. Due to its brevity, reproducing the key results are not possible yet.

Response: We apologize for our negligence on management of the GitHub. Now, all the scripts and codes for gene annotation, abundance calculation, machine learning, statistical analysis and visualization used in this study are available online at <https://github.com/ZhenyanZhang/ARG-global>. We have now provided the numerical data underlying all figures in Supplementary Data 1 (data for Figures in the main text) and Supplementary Data 2 (data for Supplementary Figures). We are trying our best to make possible the reproduction of our results so we will be very happy to conduct any additional procedure that the editor or the referee request us at this regard.

Detailed comments:

1. Line 31, "... largely driven by the host range". Please explain the meaning of "host

range”.

Response: Sorry for the unclear description. We have now deleted this sentence in the revised manuscript.

2. Line 34, “Our results demonstrated that about 25% of the ARGs posed a health risk,” based on the analysis, is the 25% arbitrarily determined by the author (see Line 227)?

Response: Our results demonstrated that 23.78% of the ARGs posed a health risk. We have now revised this sentence with the exact value (line 33).

3. Line 37, “...global health risk...”. Please explain more on this term “global health risk”

In the abstract.

Response: Sorry for the unclear description. The health risk in this study means the risk for ARGs confounding the clinic treatment for pathogens. We have added this explanation in the revised abstract (line 30-31).

4. Line 81, “..., it is the first analysis of ARGs from MAGs rather than from unassembled sequences.”. I searched “assembled metagenome genomes antibiotic resistance”, and found this manuscript analyzes ARGs detected from MAG (10.1021/acs.est.5b03522) in 2019.

Response: We thank the reviewer for pointing this out. This sentence has now been removed from the revised manuscript.

5. Line 114. “Genes conferring multidrug resistance dominated in most sub-habitats, especially in human, engineered, freshwater and terrestrial environments (Fig. 1c). These results implied that, in addition to geographic factors (Fig. S2), anthropogenic use of antibiotics influenced the composition and abundance of the ARGs.” The logic implication is not clear: as all environments have their specific geographic factors that can influence ARGs, based on the figures, it is not convincing that anthropogenic use of antibiotics has the effects to influence ARGs as well.

Response: Thanks for this useful comment for improving the readability of our article. Geographic factors like latitude was reported to influence the abundance of the ARGs¹⁰, we also confirmed this result in the study (Supplementary Fig. 2). However, anthropogenic activities are also critical for dissemination of ARGs, because genes resisted to widely used antibiotics like tetracyclines were prevalent in diverse habitats and abundant in the human-associated habitats. We then calculated population densities at each sample site to further determine the impacts of anthropogenic activities on the dissemination and abundance of ARGs (Figs. 2d,e,f). We have revised this sentence accordingly in the manuscript (line 108-112).

6. Line 140, in figure S5, only a few genes were analyzed. But at the beginning of this paragraph “One hundred and ninety-one ARGs...”. The author needs to explain the discrepancies in the number of genes.

Response: Thanks for pointing it out. We have now clarified that we only showed the

ARGs carrying out biological functions besides antibiotic resistance in Supplementary Fig. 5.

7. Line 141, “These genes, initially identified as ARGs, clearly perform biological functions other than antibiotic resistance¹⁹. I am not clear about the conclusion. It seems these ARGs can perform AR related function according to RGI annotation. I do not understand why they do not in this sentence.

Response: We apologize for the unclear description that misled the reviewer. We have now revised this sentence in manuscript as “There were 43 genes initially identified as ARGs that clearly perform biological functions in addition to antibiotic resistance.” (line 134-135). These ARGs can perform AR related function according to RGI annotation, by the alignment of nucleotide sequence. However, some of these ARGs performed other functions in their natural habitat in addition to AR¹¹. For example, ARGs with multidrug efflux pumps are ubiquitous, and their original function in nature was not to confer resistance to antibiotics that are currently used in human therapy¹²⁻¹⁴. This finding underscores the necessity of evaluating the risk of ARGs on a case-by-case basis.

8. Line 156, “This result confirms the influence of antibiotic use on the distribution of ARGs, because these are commonly used antibiotic classes.” Can authors explain whether adeF has some relationship to fluoroquinolone and tetracycline? Since adeF is used as an example here, it is necessary to examine or explain its relationship to

antibiotics.

Response: We again thank the reviewer's comment. The Comprehensive Antibiotic Resistance Database (CARD)⁶ was a database for ARG annotation with genomes and metagenomic reads in this study. It provides data, models, and algorithms relating to the molecular basis of antimicrobial resistance. As classification in CARD, *adeF* were the genes exhibiting resistance to fluroquinolone and tetracycline. However, we have now deleted this unclear description from the manuscript since all data were re-analyzed and *adeF* was not used as an example in manuscript.

9. Line 184, "ARGs could be linked to more than one MGE (Fig. S8)." Please explain how to determine the "link" between ARG and MGE.

Response: Thanks for this useful comment for improving the readability of our article. We collected 27013 completed genomes in NCBI RefSeq database⁴ to determine the links between ARG and MGE. We extracted 5 kb upstream and downstream of the ARGs detected in all completed genome and annotated them with the MGE databases. Such close proximity of MGEs and ARGs is more likely to induce HGT⁹. ISs were annotated with the ISfinder database¹⁵ using BLASTN (v0.7.13; e-value $\leq 10^{-10}$, identity $\geq 80\%$, coverage $\geq 80\%$), while the transposases and integrases were annotated with the NCBI Reference Sequence, which were clustered using CD-HIT¹⁶ with the threshold $\geq 90\%$, at an e-value $\leq 10^{-10}$ with a minimum amino acid identity of 60% over 60% query coverage using Diamond¹⁷ (v0.9.36.137). We have now added this information in the revised version of the manuscript (lines 196-198, 420-428).

10. Line 203, "...the hosts of 125 ARGs were all pathogens (human pathogenicity = 1; Table S2)." I cannot find the method descriptions on how to determine the hosts of ARGs. Table S2 does not list host either.

Response: Sorry for the missing methods. For the distribution of ARG hosts, we identified these hosts by implementing stricter quality criteria following an insightful suggestion by reviewer #2: only considering ARGs in contigs longer than 10 kb and making sure that the taxonomic affiliation of any genes found in those ARG contigs agrees with the overall taxonomy of the MAG (Fig. 2a) (line 159-163). For the determination of the mobility and human pathogenicity of ARGs, we collected 27013 completed genomes in NCBI RefSeq database⁴ (see "Methods" section; Fig. 8h) and annotated them with CARD (line 420-421). We have provided the information of ARG hosts in Supplementary Tables 4, 5 and 10.

11. Line 264, when building the machine learning model, what is the sample sizes for the training and test datasets? What is the most important predictor in the prediction model? Are the responses categorical variables (11 categories)? If so, how many samples are in each category? Can you provide ROC plots?

Response: Sorry for missing of such important information in the previous version. We used 712 samples from marine habitats to establish the prediction model (Supplementary Table 11). Random forest was an accurate algorithm by using bootstrap sample: each tree was built by about 2/3 samples of all data and model

performance was validated by the remaining out-of-bag data¹⁸. To further ensure the performance of random forest and avoid overfitting, we used the ten-fold cross-validation. The original dataset was randomly partitioned into 10 folds. In each round, nine folds were used to train the model as the training set, and the remaining data as the test set to evaluate the model¹⁹. The results of ten-fold cross-validation were evaluated by confusion matrix (Supplementary Fig. 14). We also evaluated the performance of machine learning using a ROC plot (Fig. 4c), which constructed in each rank.

The importance of each indicator was also determined by machine learning to figure out the most critical factors influencing the antibiotic resistance threats in marine habitats. Latitude, which has been confirmed to significantly influence the abundance of ARGs in this study (Supplementary Fig. 2), was the most important predictor in the prediction model (Fig. 4d). At the meantime, climate change stressors including ultraviolet radiation changes, sea level rise, surface temperature rise and ocean acidification were also critical for prediction model, and confirmed that climate change caused by anthropogenic activities could greatly influence the risk of antibiotic resistance and need to be addressed^{20,21}.

The distribution of the risks for marine samples were uneven (Supplementary Fig. 11), so for better prediction accuracy, the dataset was discretized by three unsupervised methods (k-means, equal width, and equal frequency)^{2,3}, and the samples were then divided into 10 ranks according to their risks (rank 10 for the highest risk and rank 1 for the lowest risk). The different methods discretized the

dataset in different results (Supplementary Figs. 12, 13). Equal frequency resulted in the well-distributed dataset, however, it failed to clearly distinguish the samples in ranks 1 to 5. On the contrary, equal width clearly differentiated the samples in each rank, but nearly all the samples were grouped as rank 1 and only one samples in some ranks. K-means algorithm, the most known and used clustering method², balanced the sample number (not strictly even but better than the original dataset and equal width) and dissimilarity in each rank. The number of samples as well as the minimum, maximum and interval value for each rank were shown in Supplementary Fig. 12 and 13.

More details can be found in the manuscript, and we hope we have clearly described it now.

12. Figure 2a, please explain the meaning of the circles and zoom circles.

Response: Sorry for the confused expression of this figure. We deleted it in the revised version of the manuscript.

13. Line 580, "...by Eloie-Fadrosh et al (2020)28". The citation style is not correct, as here refers to the name of the last author, but the name of the first author should be used.

Response: Sorry for the mistake. All the references have been checked in the revised version of the manuscript.

14. Line 612, "...the number of antibiotics to which the ARG conferred resistance." Please explain how the number is calculated.

Response: For example, according to the classification in CARD, *tetC* exhibited resistance to the macrolide, fluoroquinolone, aminoglycoside, carbapenem, cephalosporin (cephamycin), glycylicline, penam, tetracycline, peptide, aminocoumarin, rifamycin, phenicol, triclosan, and penem. So, the number of antibiotics that *tetC* conferred resistance is 14. And the clinic availability of *tetC* was the sum of the consumption of these 14 antibiotics. We added this information in the revised version of the manuscript (line 463-469).

15. Definition of pathogenic host is not clear. By looking at the A-to-Z database (<https://www.bode-science-center.com/center/relevant-pathogens-from-a-z.html>), I am not sure how to use it to determine pathogenic and non-pathogenic host.

A relevant question is about "We also determined the potential human pathogenicity of ARGs based on the proportion of pathogens in their hosts to evaluate the health risk of clinical ARGs.". Here, how do you determine the proportion of pathogens? Are "hosts" referring to human in this sentence?

Response: Thanks for these useful requests of clarification. The pathogenic genomes in this study were defined by comparison of their taxonomical information to the A-to-Z database which is continually updated with clinically relevant pathogens and aligned with information from the Robert Koch Institute in Germany and the WHO²² (line 380-383). For example, *Acinetobacter baumannii* is one of the pathogens in this

dataset, so all the MAGs or completed genomes taxonomically assigned as *Acinetobacter baumannii* were identified as pathogens. We provided the pathogenic identification for each genome in Supplementary Tables 4, 5 and 10. We determined the potential human pathogenicity of one ARG based on the proportion of pathogens in all hosts of this ARGs as Eq. 2 showed. For example, if one ARG is carried by 100 hosts, and 50 hosts were pathogens, then the human pathogenicity of this ARG is 50%.

References

- 1 Wikipedia. *List of countries and dependencies by population density*, https://en.wikipedia.org/wiki/List_of_countries_and_dependencies_by_population_density (15 Nov 2021).
- 2 Sinaga, K. P. & Yang, M. Unsupervised K-Means clustering algorithm. *IEEE Access* **8**, 80716-80727 (2020).
- 3 Helskyaho, H., Yu, J. & Yu, K. in *Machine Learning for Oracle Database Professionals: Deploying Model-Driven Applications and Automation Pipelines* (eds Heli Helskyaho, Jean Yu, & Kai Yu) 39-95 (Apress, 2021).
- 4 NCBI-FTP-Server. <ftp://ftp.ncbi.nlm.nih.gov/blast/db/16SMicrobial.tar.gz>. NCBI. (20 Nov 2021)
- 5 Brito, I. L. Examining horizontal gene transfer in microbial communities. *Nature Reviews Microbiology* **19**, 442-453 (2021).
- 6 Jia, B. et al. CARD 2017: expansion and model-centric curation of the comprehensive antibiotic resistance database. *Nucleic acids research* **45**, D566-D573 (2017).
- 7 Li, H. & Durbin, R. Fast and accurate short read alignment with Burrows-Wheeler transform. *Bioinformatics* **25**, 1754-1760 (2009).
- 8 Li, H. et al. The Sequence Alignment/Map format and SAMtools.

- Bioinformatics* **25**, 2078-2079 (2009).
- 9 Ellabaan, M. M. H., Munck, C., Porse, A., Imamovic, L. & Sommer, M. O. A. Forecasting the dissemination of antibiotic resistance genes across bacterial genomes. *Nature Communications* **12**, 2435 (2021).
 - 10 Bahram, M. et al. Structure and function of the global topsoil microbiome. *Nature* **560**, 233-237 (2018).
 - 11 McRose, D. L. & Newman, D. K. Redox-active antibiotics enhance phosphorus bioavailability. *Science* **371**, 1033 (2021).
 - 12 Piddock, L. J. V. Multidrug-resistance efflux pumps ? not just for resistance. *Nature Reviews Microbiology* **4**, 629-636 (2006).
 - 13 Martínez, J. L., Coque, T. M. & Baquero, F. What is a resistance gene? Ranking risk in resistomes. *Nature Reviews Microbiology* **13**, 116-123 (2015).
 - 14 Martinez, J. L. et al. Functional role of bacterial multidrug efflux pumps in microbial natural ecosystems. *FEMS Microbiology Reviews* **33**, 430-449 (2009).
 - 15 Siguiet, P., Perochon, J., Lestrade, L., Mahillon, J. & Chandler, M. ISfinder: the reference centre for bacterial insertion sequences. *Nucleic acids research* **34**, D32-D36 (2006).
 - 16 Fu, L., Niu, B., Zhu, Z., Wu, S. & Li, W. CD-HIT: accelerated for clustering the next-generation sequencing data. *Bioinformatics* **28**, 3150-3152 (2012).
 - 17 Buchfink, B., Reuter, K. & Drost, H.-G. Sensitive protein alignments at tree-of-life scale using DIAMOND. *Nature Methods* **18**, 366-368 (2021).
 - 18 Bylander, T. Estimating generalization error on two-class datasets using out-of-bag estimates. *Machine Learning* **48**, 287-297 (2002).
 - 19 Ban, Z. et al. Machine learning predicts the functional composition of the protein corona and the cellular recognition of nanoparticles. *Proceedings of the National Academy of Sciences* **117**, 10492 (2020).
 - 20 MacFadden, D. R., McGough, S. F., Fisman, D., Santillana, M. & Brownstein, J. S. Antibiotic resistance increases with local temperature. *Nature Climate Change* **8**, 510-514 (2018).

- 21 Reverter, M. et al. Aquaculture at the crossroads of global warming and antimicrobial resistance. *Nature Communications* **11**, 1870 (2020).
- 22 HARTMANN science centre. *Relevant pathogens from A-Z*, <https://www.bode-science-center.com/center/relevant-pathogens-from-a-z.html> (20 Nov 2021).

Reviewer comments, second review –

Reviewer #1 (Remarks to the Author):

The authors have addressed all my comments. I think it can be accepted with no changes.

Reviewer #2 (Remarks to the Author):

I thank the authors for their extensive restructuring of their manuscript and for now focusing their analysis at the metagenomic read level. I also commend the authors for using stricter criteria when analysing ARG host association with the MAGs. I have no further comments.

Reviewer #3 (Remarks to the Author):

Thank you for the revision and response. I can understand you better now. However, a few issues remain. Please revise accordingly.

Line 128, “The distributions of 1106 ARGs were not significantly influenced...” Can you explain where 1106 come from as I do not see this from Fig 1e or 1f.

Line 137, “... of evaluating the risk of ARGs on a case-by-case basis.” Can you elaborate more on this? I am confused about “risk” as it is not mentioned in the paragraph, and “case-by-case” as it seems to suggest reviewing each ARG distinctively (but you actually group them). Can you rephrase this sentence?

Line 209, “The mobility of ARGs was defined as the number of associated MGEs detected, ...” please provide a reference or give more explanations.

Line 782, “There were 715 and 29 ARGs significantly enriched in high- and low-intensity human activities environment”. Here the authors classified ARGs by enrichment analysis but the method part does not provide the details or thresholds.

t

REVIEWER COMMENTS

Reviewer #1 (Remarks to the Author):

The authors have addressed all my comments. I think it can be accept with no changes.

Response: Thank you for your time and efforts handling our manuscript! Your previous comments have greatly improved our work.

Reviewer #2 (Remarks to the Author):

I thank the authors for their extensive restructuring of their manuscript and for now focusing their analysis at the metagenomic read level. I also commend the authors for using stricter criteria when analysing ARG host association with the MAGs. I have no further comments.

Response: Thank you for your time and efforts handling our manuscript! We really appreciate your comments on the methods for analysis, which is sounder and more accurate for quantifying the risk of ARGs.

Reviewer #3 (Remarks to the Author):

Thank you for the revision and response. I can understand your better now. However, a few issues remain. Please revise accordingly.

Response: Thank you for your time and efforts handling our manuscript! We have carefully revised the following issues according to your comments. All your insightful comments have greatly improved our work. We really appreciate your comments!

Line 128, “The distributions of 1106 ARGs were not significantly influenced...” Can you explain where 1106 come from as I do not see this from Fig 1e or 1f.

Response: Sorry for the mistake. We totally found that 1846 ARGs shared in the area with low- or high-intensity human activities (Fig. 1e). Among them, abundance of 715 and 29 ARGs significantly increased in high- and low-intensity human activities environment, respectively (Fig. 1f). Thus, it should be 1102 ARGs were not significantly influenced. We have revised in the manuscript and added the results of two-tailed Welch’s t-test of all 1846 shared ARGs in Supplementary Data 3 (line 124-127). Besides, we re-checked all the numbers in manuscript and make sure they are right now. Thank you again for pointing this out.

Line 137, “... of evaluating the risk of ARGs on a case-by-case basis.” Can you elaborate more on this? I am confused about “risk” as it is not mentioned in the paragraph, and “case-by-case” as it seems to suggest reviewing each ARG distinctively (but you actually group them). Can you rephrase this sentence?

Response: Thank you for your useful suggestion. Results indicated that different ARGs exhibit the different level of correlation to the anthropogenic activities, which will influence the health risk of ARGs on human lives. We then quantitatively evaluate the health risk of each ARGs considering the four indicators (human accessibility, mobility, human pathogenicity and clinical availability) in the following sections. We revised this part in the manuscript as your suggestion (line 133-137).

Line 209, “The mobility of ARGs was defined as the number of associated MGEs detected, ...” please provide a reference or give more explanations.

Response: Thank you for your useful suggestion. It is now clearer than ever that MGEs were greatly responsible to the dissemination of ARGs and used for determining the mobility of ARGs in the previous studies, which assessed the health risk of ARGs qualitatively^{1,2}. In the present study, for quantitative analysis, the mobility of ARGs was defined as the number of associated MGEs detected (see “Methods” section; Supplementary Data 7). It should be noted that it is almost impossible to measure the absolute value of the mobility of ARG, which can be changed with the genetic contexts in specific species, because of the fitness costs in HGT¹. However, our method determined a potential mobility of ARGs, which was critical for risk assessment. We revised the manuscript and added more explanations according to your insightful suggestion (line 207-215).

- 1 Martínez, J. L., Coque, T. M. & Baquero, F. What is a resistance gene? Ranking risk in resistomes. *Nature Reviews Microbiology* **13**, 116-123 (2015).
- 2 Zhang, A.-N. et al. An omics-based framework for assessing the health risk of antimicrobial resistance genes. *Nature Communications* **12**, 4765 (2021).

Line 782, “There were 715 and 29 ARGs significantly enriched in high- and low-intensity human activities environment”. Here the authors classified ARGs by enrichment analysis but the method part does not provide the details or thresholds.

Response: Thank you for pointing this out. In this analysis, the average abundance of each ARG in high- and low-intensity human activities environment were compared directly, and the two-tailed Welch's t-test were used for determining the significance (p -value were adjusted by FDR). We did not use the enrichment analysis. Sorry for the unclear description, we have revised the sentence as "The abundance of 715 and 29 ARGs significantly increased in high- and low-intensity human activities environment, respectively (adjust $p < 0.05$, two-tailed Welch's t-test)". At the meantime, we added the method of FDR in the Methods section (line 523), as it was missed in the previous version. Thank you again for all the useful comments, which greatly improve the quality of our manuscript.